# Development of Polylactic Acid Thermoplastic Starch Formulations Using Maleinized Hemp Oil as Biobased Plasticizer

**DOI:** 10.3390/polym13091392

**Published:** 2021-04-25

**Authors:** Alejandro Lerma-Canto, Jaume Gomez-Caturla, María Herrero-Herrero, Daniel Garcia-Garcia, Vicent Fombuena

**Affiliations:** 1Technological Institute of Materials (ITM), Universitat Politècnica de València (UPV), Plaza Ferrándiz y Carbonell 1, 03801 Alcoy, Spain; allercan@epsa.upv.es (A.L.-C.); jaugoca@epsa.upv.es (J.G.-C.); dagarga4@epsa.upv.es (D.G.-G.); 2Centre for Biomaterials and Tissue Engineering, Universitat Politècnica de València (UPV), Camí de Vera s/n, 46022 Valencia, Spain; maherhe7@etsii.upv.es

**Keywords:** polylactic acid, thermoplastic starch, maleinization process, hempseed oil, mechanical properties, thermal properties

## Abstract

In this study, hemp seed oil was reacted with maleic anhydride in an ene reaction to obtain maleinized hemp seed oil (MHO). The use of MHO as a plasticizer and compatibilizer has been studied for polylactic acid (PLA) and thermoplastic starch (TPS) blends (80/20, respectively). By mechanical, thermal and morphological characterizations, the addition of MHO provides a dual effect, acting as plasticizer and compatibilizer between these two partially miscible biopolymers. The addition of MHO up to 7.5 phr (parts by weight of MHO per hundred parts of PLA and TPS) revealed a noticeable increase in the ductile properties, reaching an elongation at break 155% higher than the PLA/TPS blend. Furthermore, contrary to what has been observed with maleinized oils such as linseed oil, the thermal properties do not decrease significantly as a result of the plasticizing effect, due to the compatibilizing behavior of the MHO and the natural antioxidants present in the oil. Finally, a disintegration test was carried out in aerobic conditions at 58 °C, for 24 days, to demonstrate that the incorporation of the MHO, although causing a slight delay, does not impair the biodegradability of the blend, obtaining total degradation in 24 days.

## 1. Introduction

Society’s increasing ecological concerns, as well as the pollution problems arising from the indiscriminate use of plastics and the large amount of waste currently generated, have led to an increase in research into more environmentally friendly polymeric materials that are capable of replacing the commonly used petroleum-derived polymers currently in use [1].

Polylactic acid (PLA) is one of the most promising bioplastics and one of the most widely used industrially nowadays due to its biodegradability and biocompatibility, as well as its market availability and low price compared to other biopolymers. Global PLA production in 2020 was close to 380,000 tonnes, representing 18.7% of total bioplastics production, and is estimated to increase to 560,000 tonnes by 2025 [2]. PLA is a linear aliphatic polyester synthesized from the ring-opening polymerization (ROP) of lactic acid obtained from the fermentation process of sugar-rich compounds such as cellulose and starches [3]. PLA exhibits a number of properties, such as good mechanical strength, ease of processing and high transparency, that make it interesting for use in sectors such as medicine, agriculture, automotive, textile, 3D printing and packaging, among others [4]. However, it has several limitations, such as high hardness, high stiffness, high brittleness and poor thermal and barrier properties, that limit its industrial application. Numerous methodologies have been studied to improve the drawbacks of PLA, with physical blending with other more ductile polymers and plasticization being the most studied methods, as well as the most economical. Some studies have reduced the brittleness of PLA by physically mixing it with other bio-based and/or biodegradable polymers such as poly(3-hydroxybutyrate (PH3B) [5,6], poly(ε-caprolactone) (PCL) [7,8,9], polybutylene succinate-co-adipate (PBSA) [10,11] or polybutylene adipate-co-terephthalate (PBAT) [12,13,14]. One of the most interesting and well-studied biopolymers used in physical blends to reduce the drawbacks of PLA is thermoplastic starch (TPS) [15,16,17]. TPS is a highly biodegradable, inexpensive and abundantly available polymer. Moreover, this biopolymer is characterized by high flexibility, which is interesting to increase the flexibility and elongation at break of PLA [18]. However, one of the main problems with the physical blending of PLA/TPS is the lack of miscibility between both polymers, which limits the improvement of the resultant formulations [19]. Many studies have demonstrated the effectiveness of the use of compatibilizers such as maleic anhydride (MA) [20,21] or polyethylene glycol (PEG) [15,22] to improve the interfacial adhesion between the two polymers. On the other hand, the scientific literature contains a great number of studies about the use of different plasticizers for PLA that have improved its ductile mechanical properties, such as polyethylene glycol (PEG) [23,24], acetyl tributyl citrate (ATBC) [25,26] or oligomeric lactic acid (OLA) [27,28]. However, the use of vegetable oils (VOs) as plasticizers for PLA has gained importance in recent years due to their multiple benefits: they are easy to obtain, biodegradable, from renewable sources, highly available, non-toxic and their migration is lower than that of other plasticizers [29]. 

Hemp (*Cannabis Sativa* L.) is an annual herbaceous plant, which has been cultivated for many centuries as a source of food, for its fibers and for its oil [30]. Hemp seeds are characterized by their high oil yield, which is between 28% and 35% depending on the variety, geographical region of cultivation or climatic conditions [31]. Hemp seed oil (HO) is a product of great interest due to its nutritional and health benefits. HO has been shown to reduce cholesterol and blood pressure as well as prevent cardiovascular diseases and cancers [32]. Moreover, HO can be easily obtained by cold-pressing the seeds or using organic solvents [33] and is characterized by a high content of linoleic (n-6) and α-linolenic (n-3) fatty acids [34]. Due to its high unsaturation content, the HO allows a wide variety of chemical modifications, such as epoxidation, acrylation, hydroxylation or maleinization [35,36]. These modifications increase the vegetable oil’s reactivity by improving its interaction with the polymeric matrix, giving it a greater plasticizing effect than unmodified vegetable oil. The increased oil reactivity after chemical modification improves the penetration of the plasticizer into the matrix, leading to an increase in the intermolecular distance between the polymer chains and therefore an increase in the free volume. This three-dimensional space is occupied by the oil itself, increasing the space between the polymer chains and reducing the intensity of the interactions between them. Thus, modified vegetable oils improve the mobility and therefore their ductile mechanical properties as well as facilitating their processing [37]. This makes modified vegetable oils a great alternative to petroleum-based plasticizers. Different modified vegetable oils, such as epoxidized vegetable oils (soybean oil (ESBO) [38], epoxidized linseed oil (ELO) [39], epoxidized karanja oil (EKO) [40], epoxidized jatropha oil (EJO) [41], epoxidized palm oil (EPO) [42] and maleinized vegetable oils (maleinized linseed oil (MLO) [37,43], maleinized cottonseed oil (MCSO) [43]) have been successfully used as plasticizers for PLA, improving its ductile mechanical properties and increasing the environmental efficiency of the formulations obtained. It has also been shown that modified vegetable oils can act as compatibilizers in physical polymer blends as they can react with both polymers, leading to a chemical bridge between them, thus improving their compatibility. In this sense, Ferri et al. [37] demonstrated that the MLO, in addition to plasticizing, improved the compatibility of the PLA/TPS (70/30) blend due to the fact that the maleic anhydride groups of the MLO are able to react with the hydroxyl groups present in both PLA and TPS, thus improving the compatibility between them. The improved compatibilization between both polymers was reflected in an increase in the ductile mechanical properties, such as elongation at break, observing how the incorporation of 6 phr of MLO improved the elongation at break by around 645% with respect to the unplasticized blend. A similar trend was also observed for impact energy absorption. Przybytek et al. [44] observed that ESO acted as a plasticizer and compatibilizer in PLA/TPS blends (75/25), improving their interfacial adhesion and ductile properties with respect to the unplasticized blend. In this case, the authors observed that the incorporation of 2 phr of ESO increased the elongation at break and the impact absorption energy by 82% and 20%, respectively, regarding the unplasticized PLA/TPS blend. Turco et al. [45] also observed an improvement in the interfacial adhesion between PLA and TPS after the addition of cardoon seed epoxidized oil.

The aim of this work has been to study the potential of this maleinized oil as a green plasticizer and/or compatibilizer in PLA/TPS (80/20) blends. Specifically, in this case, the effect of the incorporation of different amounts of maleinized hemp seed oil on the mechanical, thermal and morphological properties of PLA/TPS (70/30) blends has been studied.

## 2. Materials and Methods

### 2.1. Materials

PLA commercial-grade Ingeo Biopolymer 2003D was supplied by NatureWorks LLC (Minnetonka, MN, USA). This PLA was characterized by a density of 1.24 g·cm^−3^ and a melt flow index range of 6 g·10 min^−1^ at a temperature of 210 °C and a load of 2.16 kg. TPS with a commercial-grade Mater-Bi®NF 866 was supplied by Novamont SPA (Novara, Italy). In this case, TPS was characterized by a density of 1.27 g·cm^−3^ and melt flow rate of 3.5 g·10 min^−1^ at 150 °C. Melting peak temperature was obtained at 115 °C. In addition, hemp seed oil (Cannabis sativa), used as green compatibilizer–plasticizer, was extracted using a CZR-309 press machine (Changyouxin Trading Co., Zhucheng, China) at room temperature. Seeds of Cannabis sativa were obtained from a local market in Callosa de Segura, Spain. Extracted oil was characterized by an acid value of 100–115 mg KOH·g^−1^ and viscosity of 10 dPa·s at 20 °C. Maleic anhydride (MA), with a purity higher than 98%, used in the maleinization process, was supplied by Sigma Aldrich (Madrid, Spain).

### 2.2. Maleinization of Hemp Seed Oil

To synthesize MHO, a three-neck round flask with a capacity of 500 mL, equipped with a heating mantle, was used. A propeller mechanical stirrer was connected to the central neck, a thermometer was connected in the second neck, and the third neck was used to add MA and allowed sample extraction for the evaluation process. The experimental process was carried out for 3 h as follows: first, 300 g of HO was introduced in the round flask, maintained with constant stirring (200 rpm) and heated up until a temperature of 180 °C was reached. In this first stage, 9 g of MA per 100 g of HO was dropped into the flask and subjected to constant agitation for 1 hour. Afterward, the process was replicated at 200 °C and 220 °C, adding 9 g of MA per 100 g of HO at every stage. Therefore, the ratio used was 2.4/1, in line with Carbonell-Verdu et al. [3] in previous reports. To evaluate the extent of the maleinization process, samples were extracted every 30 min. Finally, MHO was cooled down to room temperature and centrifuged at 4000 rpm to purification. Figure 1 shows a schematic representation of the process developed.

To evaluate the maleinization process, the acid number was obtained following the equation indicated in the ISO 660:2009: (1)$$\mathrm{Acid}\;\mathrm{value}=\frac{56.1\times \;V\;\times \;C}{m}$$
where C is the exact concentration of KOH standard solution (mol·L^−1^), V is the volume of KOH used to titrate the sample (mL) and m is the mass of maleinized oil used to titrate it (g).

### 2.3. Processing and Compatibilization of PLA/TPS 

Prior to processing, PLA and TPS pellets were dried in an air circulating oven at 60 °C for 24 h to remove moisture. A fixed weight content of PLA and TPS of 80 wt.% and 20 wt.%, respectively, was set while MHO, used as compatibilizer and plasticizer, was added from 2.5 to 10 parts per hundred resin (phr). Table 1 summarizes the different compositions developed. The percentages of oil were selected due to previous works obtained with maleinized vegetable oil, in which amounts higher than 10% of maleinized vegetable oils showed signs of saturation [37]. After the drying process, polymeric samples were mechanically mixed in a zipped bag with the corresponding amount of MHO. Mixtures were extruded in a twin-screw co-rotating extruder at a rotating speed of 40 rpm and selected temperatures of 162.5 °C, 165 °C, 170 °C and 175 °C from the hopper to the die. Extruded mixtures were air-cooled to room temperature and pelletized prior to processing by injection molding in a Meteor 270/75 from Mateu & Solé (Barcelona, Spain) at an injection temperature of 175 °C. Samples obtained were in accordance with tensile test standards; they were rectangular samples with a size of 80 × 10 × 4 mm^3^ for the subsequent thermal and mechanical characterization. 

### 2.4. Mechanical and Thermal Characterization 

Tensile properties of the different compounds developed, as a minimum of ten samples per compound, were measured by an electromechanic universal test machine, Ibertest Elib 30 (Ibertest S. A. E, Madrid, Spain), according to ISO 527. The tensile samples had a weightlifting shape and a total length of 150 mm. The narrow part was 6 mm in length and the breadth was 10 mm. The load cell used was 5 kN and the crosshead speed was set to 10 mm·min^−1^. Impact energy of unnotched samples was carried out following the guidelines of ISO 179, using a 6 J Charpy pendulum Metrotec S. A. (San Sebastian, Spain). A minimum of five samples were analyzed with sizes of 80 × 10 × 4 mm^3^. The Shore D hardness of PLA/TPS compounds was obtained using a Shore D hardness durometer 676-D (J. Bot. S. A., Barcelona, Spain) according to ISO 868. Five samples were tested, and average values are shown. 

In order to analyze the thermal properties of the obtained blends, a complete characterization was carried out. The calorimetric analysis was performed using a DSC Mettler-Toledo 821e (Schwerzenbach, Switzerland). Under a nitrogen atmosphere with a flow rate of 66 mL·min^−1^, specimens were subjected to heating (30–350 °C at rate of 10 °C·min^−1^). The crystallinity of the compounds was calculated using the following equation: (2)$$X_{c}\left(\%\right)=\frac{\Delta H_{m}-\Delta H_{c}\;}{w\cdot \Delta H_{m}^{o}}\times 100$$
where w refers to the weight fraction corresponding to PLA. ΔH_m_ is the melting enthalpy (J·g^−1^), ΔH_c_ represents cold crystallization (J·g^−1^) and $\Delta H_{m}^{o}$ represents the melting enthalpy for a PLA structure that is theoretically fully crystalline, 93 J·g^−1^, as reported in the literature [46]. 

Thermogravimetric analysis (TGA) was carried out to analyze the weight loss as a consequence of the degradation process as a function of the temperature. The heating program was from 30 to 700 °C, with a nitrogen atmosphere of 66 mL min^−1^ and with a heating rate of 10 °C·min^−1^ on a TGA/SDTA851 thermobalance from Mettler Toledo Inc (Schwerzenbach, Switzerland). The onset of degradation temperature (T_0_) was assumed as the temperature at which the sample reached a 5% weight loss. The maximum degradation rate temperature (T_max_) was obtained from the corresponding peak in the first derivative curve (DTG).

Finally, dynamic mechanical thermal analysis (DMTA) in torsion mode was performed using an AR G2 oscillating rheometer from TA Instruments (New Castle, USA). Rectangular samples 40 × 10 × 4 mm^3^ in size were subjected to a temperature ramp from 30 °C up to 130 °C with a heating rate of 2 °C min^−1^ at a frequency of 1 Hz and using as a controlled variable the % strain in 0.1.

### 2.5. Morphological Characterization 

Fractured surfaces from the impact test of PLA/TPS samples were observed with a field emission scanning electron microscope (FESEM), the ZEISS ULTRA model from Oxford Instruments (Abingdon, UK). Before observation, samples were coated with a thin gold–platinum layer with a sputter-coater EM MED020 from Leica Microsystems (Wetzlar, Germany). Then, samples were observed working at an acceleration voltage of 2 kV.

### 2.6. Disintegration under Composting Conditions

The disintegration test was conducted according to the ISO 20,200 standard in aerobic conditions at 58 °C and at a relative humidity of 55%. Samples with sizes of 25 × 25 × 1 mm^3^ were placed in a carrier bag and buried in controlled soil. Previously, samples were dried at 40 °C for 24 h to remove moisture and weighed. The soil into which the samples were buried was composed of saccharose, specific feed for rabbits, urea, corn starch, sawdust, corn oil and mature compost in the proportions indicated in the ISO 20200. The disintegration process was extended for 24 days and samples were periodically unburied at 6, 10, 16, 18, 20, 22 and 24 days, washed with distilled water and placed in an air oven at 40 °C for 24 h before weighing. All tests were carried out in triplicate to ensure reliability. The average disintegration percentage of extracted samples was calculated using the following equation: (3)$$W_{l}\;\left(\%\right)=\frac{w_{0}-w}{w_{0}}\cdot 100$$
where $w_{0}$ refers to the initial dry weight of the sample and w is the weight of the sample extracted from compost soil on different days after drying. Furthermore, optical images were taken to record the progression of disintegration over time.

## 3. Results and Discussion

### 3.1. Synthesis of Maleinized Hemp Seed Oil

The acid value along the three temperature steps (180 °C, 200 °C and 220 °C) employed to study the maleinization process is graphically plotted in Figure 2. At the beginning of the reaction, HO has an acid number of 8 mg KOH g^−1^. After one hour of processing at 180 °C, a significant change in this value was obtained, reaching values around 50 mg KOH g^−1^, indicating the initiation of the maleinization process. After the second hour of the maleinization process, with a temperature of 200 °C, the acid value increased to 80 mg KOH g^−1^. As Rheinecku and Khoe [47] report in the case of vegetable oils, the maleinization reaction requires at least temperatures close to 200 °C. However, it is at the third selected temperature, 220 °C, that the results show a drastic increase, reaching values close to 105 mg KOH g^−1^. This positive increase in the acid value is due to the high availability of maleic anhydride, which can be attached to an allylic position, and also due to the ene reaction that takes place, resulting in conjugation of the two double bonds, to give the trans–trans isomer [48]. This adduct undergoes a Diels–Alder reaction with another molecule of maleic anhydride, as Teeter et al. [49] reported. On the other hand, as the reaction time increases at a constant temperature of 220 °C, the acid value tends to stabilize, giving a value of 106 mg KOH g^−1^. These values are in full agreement with those reported in the literature. For example, Ferri et al. [43] used a commercial maleinized linseed oil, characterized by acid number values from 105 to 130 mg KOH g^−1^. Therefore, the MHO developed shows similar characteristics to the few commercially available maleinized oils.

### 3.2. Mechanical Properties of PLA/TPS Blends with MHO

Unblended PLA is a rigid material with a high Young’s modulus and tensile strength (3600 MPa and 64 MPa, respectively), but it is quite brittle, with an elongation at break below 7% [50]. The addition of polymers with high ductility and flexibility, such as TPS, can improve the ductility, but the miscibility among them is a key factor to be studied. Regarding mechanical properties, it is possible to observe in Figure 3a,b that when blending PLA with 20 wt.% of TPS, a Young´s modulus of less than 2400 MPa is obtained, and the tensile strength drops to 29 MPa. Nevertheless, this decrease in mechanical properties does not result in an increase in ductile properties, as can be seen in Figure 3c,d in the Charpy´s impact resistance. The elongation at break remains at very similar values, around 7%, while the impact energy drops to 5 kJ·m^−2^. This is due to the lack of miscibility between the polymer matrices, since it is well known that PLA and starch are not chemically compatible and starch granules might make the PLA even more brittle. For this reason, the addition of a compatibilizer/plasticizer such as MHO is required in different percentages. 

With the addition of only 2.5 phr of MHO, all properties of the PLA/TPS blend changed. The Young´s modulus and the tensile strength decreased by 17.2% and 12.3%, respectively. The higher MHO content (10 phr) provides drops in the Young´s modulus and tensile strength of 49% and 24%, respectively. Thus, it is evident that mechanical resistance properties decrease when the MHO content is increased. With respect to elongation at break, the addition of MHO provides an improvement in the values obtained, a drastic change being observed with the addition of 7.5 phr (155% higher with respect to the PLA/TPS blend). It should be noted that the addition of 10 phr of MHO does not provide higher values of elongation at break, which could indicate that an optimum content of MHO could be obtained for values close to 7.5 phr. Analogously, the Charpy´s impact resistance plotted in Figure 3d gives lower values for the sample with 10 phr than for the sample with 7.5 phr. For the sample with 7.5 phr of MHO, the maximum Charpy´s impact resistance value was obtained, being 65% higher than that of the sample without MHO.

Thus, the addition of MHO provides a dual effect, with clear evidence of plasticization and compatibilization effects, turning the PLA/TPS blend from a rigid to a ductile state. Different previous studies, using vegetable oils modified by epoxidation, have obtained similar values [50,51]. The added modified oils could be situated on the surfaces of TPS particles and form a flexible layer via the reaction between the functional oil/TPS compound and the PLA matrix. In our case, this functional reactivity is achieved by the maleic anhydride present in MHO, which can react with some hydroxyl groups in PLA end chains, providing a chain extension effect. Ferri et al. [43] studied the effect of the addition of MLO, obtaining very similar values of the elongation at break (140–160%) with the addition of 8 phr of MLO. 

### 3.3. Morphological Characterization 

The fractured surfaces of the PLA/TPS blends without and with different contents of MHO were examined by SEM and the results are shown in Figure 4. Figure 4a shows a characteristic brittle facture with a smooth fracture surface and low roughness due to the very low plastic deformation achieved. Blending PLA with 20 wt.% of TPS led to signs of phase separation between PLA, acting as a polymeric matrix, and TPS, acting as a dispersed component. One of the main drawbacks of using TPS to blend PLA is that TPS is a hydrophilic polymer, whilst PLA is hydrophobic [52]. This different polarity is responsible for the lack of affinity among them. The presence of TPS results in spherical shapes dispersed throughout the PLA matrix. For this reason, and in total concordance with the mechanical results, it is necessary to implement a compatibilization strategy in order to obtain optimized blends [52]. Employing MHO as a compatibilizer/plasticizer component, morphological changes could be observed, as shown in Figure 4b. With the addition of 2.5 phr of MHO, the spherically dispersed TPS phase is more difficult to observe, indicating an improvement in the compatibility. These results may be caused by the maleic group on MHO, which could react with the hydroxyl groups present on TPS and PLA acting as chemical bonds. The plasticizing effect can be attributed to the increase in free volume caused by the insertion of MHO. In this way, the polymer–polymer interactions could be reduced and higher mobility of the polymer chains, due to the presence of long fatty acid chains in MHO, could be achieved [53], increasing the ductile properties, as illustrated in Figure 3. Figure 4c is characterized by the presence of some spherical domains corresponding to the rich TPS phase and the appearance of filaments characteristic of a more ductile breaking process with material removal. In Figure 4d, obtained with a blend with 7.5 phr of MHO, these spherical voids are less visible, obtaining a homogenous surface. Obviously, the assimilation of MHO on the PLA/TPS blends via the reaction among them leads to their saturation after the addition of a certain amount of MHO. As has been verified before, the optimal mechanical ductile properties are obtained or the sample with 7.5 phr of MHO. For this reason, Figure 4e stands out for the presence of microvoids, possibly due to an excess of MHO molecules, which provide droplets in the PLA matrix [37]. 

### 3.4. Thermal Characterization 

The thermal decomposition of the PLA/TPS blend unplasticized and plasticized with different amounts of MHO content was studied by thermogravimetry (TGA). Figure 5a shows the weight loss versus temperature curves of different samples (TG), and their corresponding first derivative curves (DTG) are plotted in Figure 5b. Additionally, Table 2 shows the most relevant degradation values, such as the degradation onset temperature (T_0_), which has been defined as the temperature at which a weight loss of 5% occurs, and the maximum degradation temperature (T_max_) obtained from the peak of the first derivative curves. As can be seen, the unplasticized PLA/TPS blend shows a T_0_ of 333.0 °C and a T_max_ of 370.3 °C. The addition of MHO in the PLA/TPS blend results in a decrease in T_0_, with this decrease being greater as the amount of plasticizer in the blend increases. In this case, the addition of 10 phr of MHO reduces the T_0_ by 10 °C with respect to the unplasticized blend. Regarding the maximum degradation of the PLA/TPS blend, the addition of different MHO amounts has a weak effect on this temperature, yielding very similar values to that of the unplasticized blend for all the samples, at around 370 °C. A similar behavior was observed by Carbonell-Verdu et al. [13] for PLA/PBAT blends (80/20) plasticized with epoxidized and maleinized cottonseed oil, noting that high plasticizer content (7.5 wt.%) resulted in a reduction in T_0_ without significantly modifying the T_max_. 

Table 2 shows the main thermal parameters, such as glass transition temperature (T_g_), melting temperature (T_m_), cold crystallization temperature (T_cc_), melting enthalpy (ΔH_m_) and crystallization enthalpy (ΔH_cc_), obtained by differential scanning calorimetry (DSC) of the PLA/TPS blends unplasticized and plasticized with different amounts of MHO content. As can be seen, the unplasticized PLA/TPS blend has a glass transition temperature (T_g_) of 61.7 °C, a melting temperature (T_m_) of around 152.5 °C and a cold crystallization temperature (T_cc_) of 123.5 °C. After the addition of the plasticizer, the melting temperature and glass transition temperature of the blend decrease slightly, with a higher decrease observed in the sample plasticized with low MHO content (2.5 phr), where a T_m_ near 148 °C and a T_g_ around 59 °C are achieved. Furthermore, it is also observed that the T_cc_ slightly decreases with respect to the unplasticized blend after the addition of different amounts of MHO content. This slight decrease in T_g_, T_cc_ and T_m_ in the samples plasticized with MHO evidences the plasticizing effect of the oil, which leads to an increase in the mobility of the polymer chains thanks to the lubricating effect, as well as to a reduction in the intensity of the interactions between the polymer chains [37,44]. However, the reported decrease is much lower than that observed in the published literature. It is here that the large amount of antioxidant compounds present in the hemp seed, and, consequently, in the oil, may play a differentiating role. Recent studies show that hemp (*Cannabis Sativa* L.) derivatives may contain more than 133 different species of cannabinoids and terpenes [54,55]. In addition, as Pollastro et al. [56] demonstrated, other substances, even if present in low amounts, such as flavonoids, stilbenoid derivatives and lingnamides, can also provide antioxidant effects. All these compounds can contribute significantly to thermally stabilizing the properties of PLA/TPS blends with different MHO contents and, despite having a plasticizing effect, as demonstrated by the mechanical properties (Figure 3), do not result in a significant decrease in the thermal properties.

Regarding crystallinity, it can be observed that the unplasticized PLA/TPS blend has a crystallinity of 3.1%. After the addition of MHO in the blend, the crystallinity increases, becoming higher as the MHO content increases. In this case, as can be seen in Table 2, the PLA/TPS blend plasticized with 10 phr MHO has the highest crystallinity, with 14.2%. This higher crystallinity in the plasticized samples is due to the plasticizing effect of the MHO, which leads to an increase in the polymeric chains’ mobility, thus facilitating the laminar rearrangement of the amorphous regions of the PLA in the blend [50,57,58].

Figure 6 shows the evolution of the storage modulus (G′) as a function of the temperature obtained by DMTA in torsion mode for PLA/TPS blends with different MHO content. With respect to the G´, representative of the elastic behavior, at temperatures below T_g_, the PLA/TPS sample shows the highest value. Nevertheless, the G´ values tend to decrease very slightly, almost negligibly, as the MHO content increases. As shown previously in Table 2, DSC revealed an almost constant T_g_ regardless of the amount of MHO introduced. Other similar studies using MLO for the plasticization of PLA and TPS blends have shown different results. In these cases, a clear decrease in the T_g_ values and consequently in the G′ value was observed as the MLO content increased [37]. Although the previous analysis of the mechanical and morphological properties demonstrated the plasticizing effect caused by the addition of the MHO, the thermal properties observed by DSC and the thermomechanical properties observed by DTMA do not show a pronounced decrease in the main parameters, such as T_g_ and G′. The maleic anhydride present in MHO is highly reactive with the hydroxyl groups present in both PLA and TPS to form ester linkages [59]. On the other hand, weak bonds can form via hydrogen bonding because of the hydrolyzation of the anhydride during the maleinization process. This bonding could link the carbonyls present in PLA and the hydroxyls present in TPS [60]. This reactive process was described by Zhang and Sun [61] with the formation of free radicals in the PLA molecule as a consequence of the addition of a catalyst. In the present study, during the extrusion process at 175 °C, such reaction-initiating radicals can be formed from the ring opening of the maleic anhydride present in the MHO. Therefore, the MHO presents a dual behavior, plasticizing the PLA and TPS blends on one side but, due to these chemical interactions, the thermal and thermomechanical properties are able to remain at similar values to those of the samples without MHO.

Thus, both the glass transition and cold crystallization processes, clearly detected by DMTA (Figure 6), show very similar curves. All samples are characterized by a G′ drop of two orders of magnitude in the 45–75 °C range. By observing the curve corresponding to the evolution of G′ for the PLA/TPS_7.5-MHO sample, it is possible to determine the best ductile properties with the lowest G′ values. On the other hand, over 7.5 phr of MHO, a slight decrease in the flexibility properties was detected, with the highest G′ values. This trend could be directly related to the saturation effect of the plasticizer, as previously noted in the mechanical (Figure 3) and morphological analyses (Figure 4) and in total concordance with other studies [62]. The cold crystallization process could be detected in the DTMA assay by the increase in the G′ from 80 to 95 °C, leading to a stiffer material. All analyzed samples show a cold crystallization process in the same range, but for the PLA/TPS blend, it is possible to observe how this crystallization temperature appears slightly earlier than in blends with MHO. This behavior is in accordance with the above-mentioned behavior observed by DSC analysis. 

### 3.5. Disintegration under Composting Conditions

Figure 7 and Figure 8 provide different images corresponding to the different disintegration times of unplasticized/uncompatibilized PLA/TPS blends and PLA/TPS blends with the addition of different percentages of MHO and the weight loss with respect to the initial mass, respectively. When the first samples were unburied from the reactor after 6 days, a noticeable visual change was detected. Samples became whitish and showed a certain surface roughness, due to the increase in crystallinity during these first stages of the disintegration process [63]. It is worth highlighting that the test was carried out under thermophilic conditions at 58 °C and a controlled humidity percentage of 55%. The proximity of the T_g_, as summarized in Table 2, increased the chains’ mobility, inducing a rearrangement, which affected the crystallinity of the samples. This visual change is in full agreement with that obtained by thermal analysis performed by DSC (Table 2). The sample with the highest degradation rate was the PLA/TPS, which showed the first signs of breakage after 16 days of incubation and reached a weight loss of more than 90% after 20 days buried under composting conditions. The addition of MHO content reduced the biodegradation rate of the PLA/TPS blends. For example, after 20 days under composting conditions, the PLA/TPS_7.5-MHO and PLA/TPS_10-MHO samples lost 60–70% of their initial weight, 38% and 25% less than the PLA/TPS blend, respectively. As mentioned above, in Table 2, the thermal analysis results demonstrate that an increase in the percentage of crystallinity was obtained with the addition of MHO, and this increase is directly related to the amount of MHO. Some authors have concluded that the disintegration process affects amorphous structures more quickly than crystalline structures, due to the difficulty for microorganisms, such as lipases, proteases and esterases, to attack this domain [64,65]. On the other hand, Thakore et al. [66] described the importance of the different compost soils, which generally contain different types of microorganisms, and concluded that, in the case of the TPS, the degradation was mainly caused by the cleavage of the ester bond due to the release of free phthalic acid and starch, while amylase acts on starch to produce reducing sugars. However, despite this disintegration-retarding effect caused by the MHO, the samples with the highest content reached 90% of weight loss at 24 days. Therefore, PLA/TPS blends compatibilized and plasticized with MHO, according to ISO 20200, can be considered biodegradable. Przybytek et al. [44] obtained similar values for disintegration over 30 days with the addition of epoxidized soybean oil (ESO) in PLA/TPS blends.

## 4. Conclusions

MHO was introduced into PLA/TPS blends to investigate its effect as a bio-based plasticizer/compatibilizer. The elongation at break of PLA/TPS is quite low, at 5%, and with the addition of PLA/TPS_7.5-MHO, an improvement of 155% was achieved. On the other hand, at values higher than 7.5 phr of MHO, a decrease in the elongation at break was observed, which led to the conclusion that there were signs of saturation in the blends. Regarding the absorbed impact energy, a 65% higher value was achieved with the addition of 7.5 phr of MHO compared to PLA/TPS without MHO, and, consequently, a decrease in hardness. From these results, it may be concluded that MHO provides an improvement in the mobility of the PLA/TPS chains and an increase in the free volume, increasing the crystallinity degree. The morphological analyses of the breakage surfaces show spherical domains belonging to TPS, since it is partially miscible in PLA. The addition of MHO improves the compatibility between both domains, as has been proven by the difficulty of observation of these spherical domains and the appearance of filaments as a consequence of ductile breakage. On the other hand, thermal parameters such as Tg or thermomechanical parameters such as G′, although they decrease slightly, do so much less sharply than in other maleinized vegetable oils. Thus, the use of MHO provides a stabilizing effect on thermal properties, despite having achieved a plasticization of the blends. This thermal stability is contributed, on the one hand, by the increase in the chemical interactions between the maleic groups and the hydroxyl groups of PLA and TPS and, on the other hand, by the high amount of antioxidant components present in MHO. Finally, disintegration under the composting conditions test showed that the addition of MHO, although it slightly delayed the process, did not result in a loss of the biodegradable capacity of the blend. Therefore, MHO is shown to be a potential plasticizer and compatibilizer of organic origin, to be used in different polymeric blends without affecting their biodegradability.

## Figures and Tables

**Figure 1 polymers-13-01392-f001:**
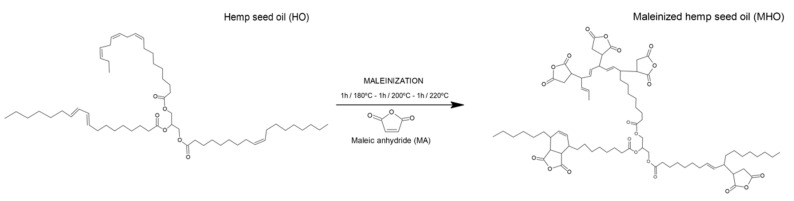
Schematic representation of the obtention of maleinized hemp seed oil (MHO).

**Figure 2 polymers-13-01392-f002:**
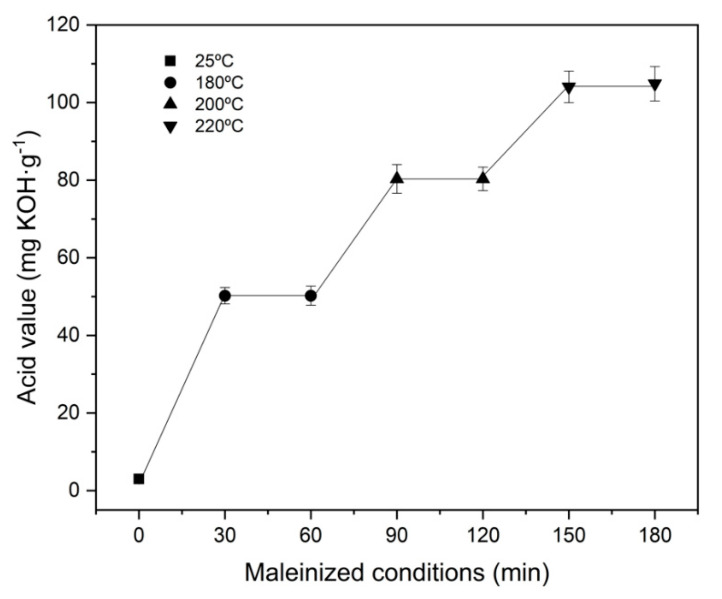
Effect of temperature and time on the efficiency of the maleinization.

**Figure 3 polymers-13-01392-f003:**
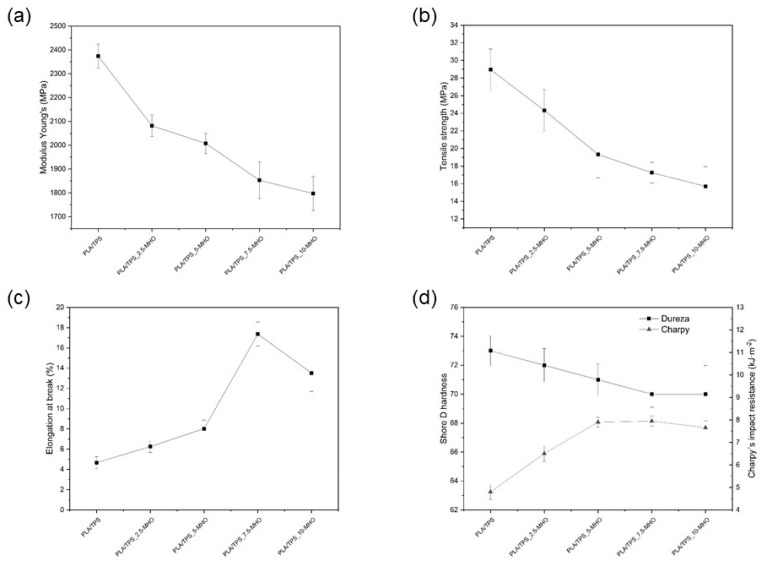
Mechanical properties of PLA/TPS with different percentages of MHO: (**a**) Young´s modulus, (**b**) tensile strength, (**c**) elongation at break, (**d**) Shore D impact and Charpy´s impact resistance.

**Figure 4 polymers-13-01392-f004:**
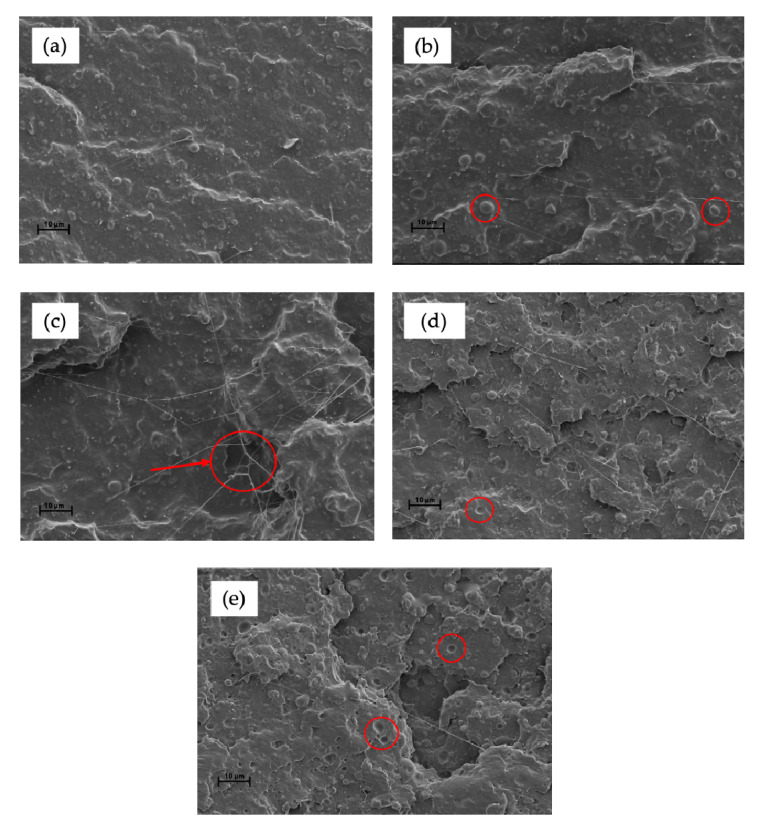
FESEM images (1000×) of the cross-section of the fractured PLA/TPS samples with MHO: (**a**) PLA/TPS; (**b**) PLA/TPS_2.5-MHO; (**c**) PLA/TPS_5.0-MHO; (**d**) PLA/TPS_7.5-MHO and (**e**) PLA/TPS_10-MHO.

**Figure 5 polymers-13-01392-f005:**
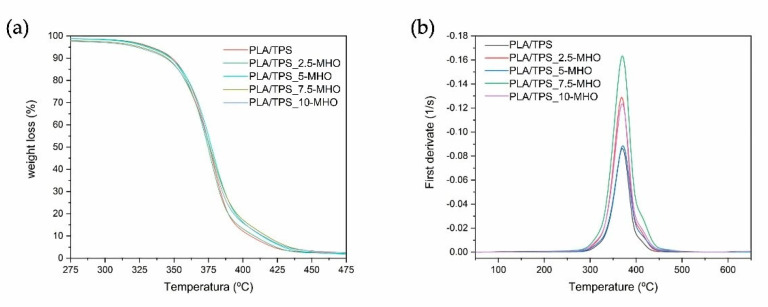
Thermal decomposition of PLA/TPS blend unplasticized and plasticized with different amounts of MHO content: (**a**) thermogravimetry (TG) weight loss and (**b**) differential thermogravimetry (DTG) first derivative curves.

**Figure 6 polymers-13-01392-f006:**
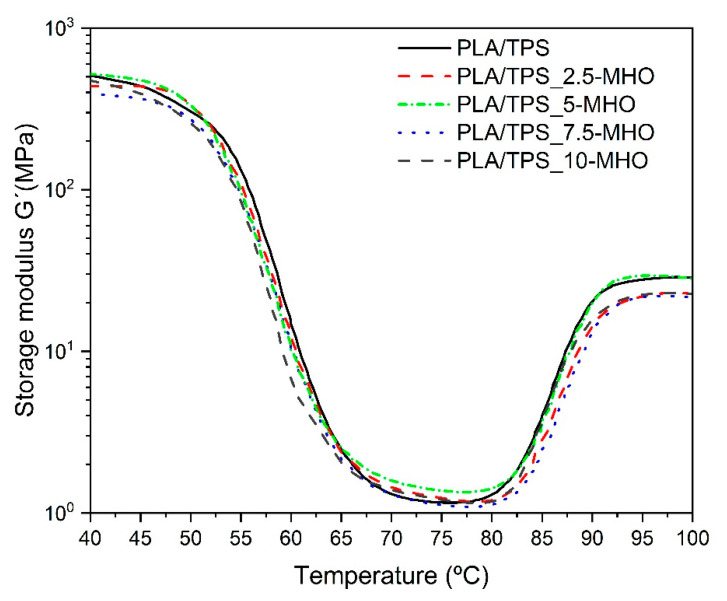
Storage module for of PLA/TPS blend unplasticized and plasticized with different amounts of MHO content as a function of temperature.

**Figure 7 polymers-13-01392-f007:**
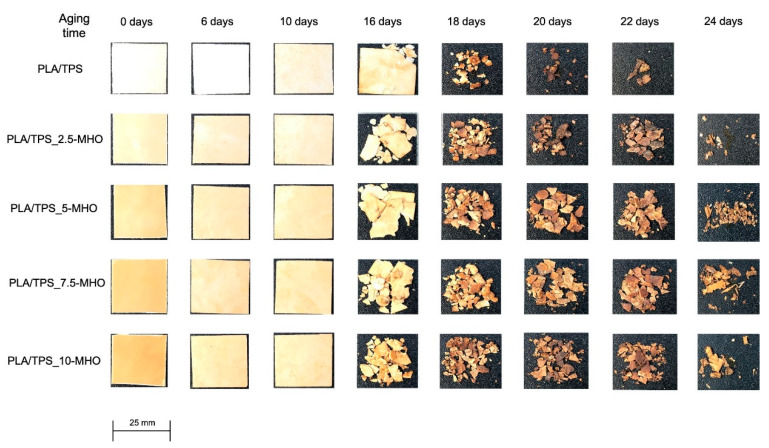
Visual appearance of disintegration in controlled compost soil of PLA/TPS blend unplasticized and plasticized with different amounts of MHO content.

**Figure 8 polymers-13-01392-f008:**
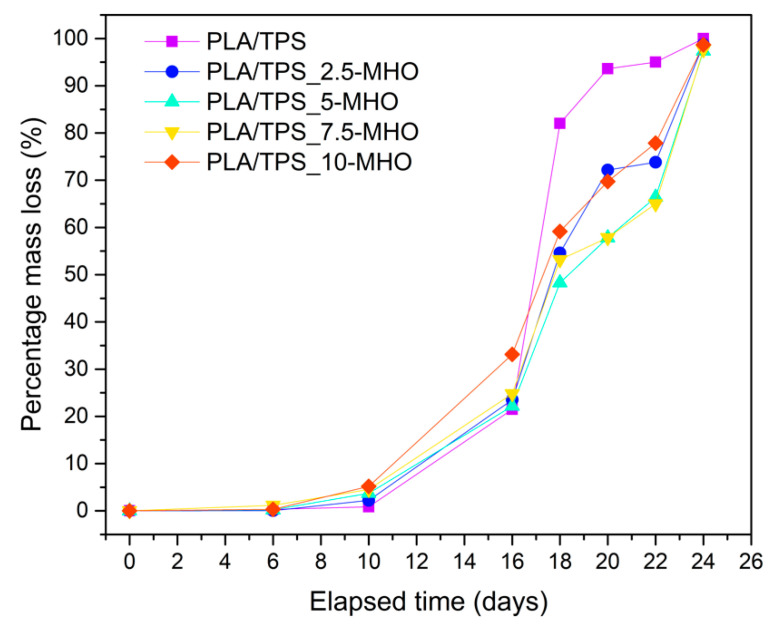
Weight loss recorded during disintegration test under composting conditions of PLA/TPS blend unplasticized and plasticized with different amounts of MHO content.

**Table 1 polymers-13-01392-t001:** Compositions of the PLA/TPS blends with varying amounts of MHO.

Code	Parts by Weight (wt. %)		MHO (phr)
	PLA	TPS	
PLA/TPS	80	20	-
PLA/TPS_2.5-MHO	80	20	2.5
PLA/TPS_5.0-MHO	80	20	5.0
PLA/TPS_7.5-MHO	80	20	7.5
PLA/TPS_10-MHO	80	20	10.0

**Table 2 polymers-13-01392-t002:** Summary of the TGA and DSC thermal parameters of PLA/TPS blend unplasticized and plasticized with different amounts of MHO content.

Code	TGA Parameters		DSC Parameters					
	T_0_ (°C)	T_max_ (°C)	T_g_ (°C)	T_m_ (°C)	ΔH_m_ (J·g^−1^)	T_cc_ (°C)	ΔH_cc_ (J·g^−1^)	Xc (%)
PLA/TPS	333	370.3	60.3	152.5	14.5	123.6	12.6	3.1
PLA/TPS_2.5-MHO	332	368.7	59.4	148.1	21.8	117.8	19.3	3.8
PLA/TPS_5.0-MHO	330	371.3	59.6	148.3	19.8	119.9	16.6	5.0
PLA/TPS_7.5-MHO	325	370.0	60.1	150.5	12.9	121.8	7.4	9.0
PLA/TPS_10-MHO	323	369.7	59.6	149.8	16.0	123.1	7.6	14.2

## Data Availability

The data presented in this study are available on request from the corresponding author.

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
