# Peer review of "Development of Polylactic Acid Thermoplastic Starch Formulations Using Maleinized Hemp Oil as Biobased Plasticizer"

_polymers, 2021, doi:10.3390/polym13091392_

Round 1

Reviewer 1 Report

Comments on polymers-1200581

The paper reports results obtained in the field of bioplastic, with the aim to improve technological properties. 
The topic is interesting and nowadays important. Experiments are well designed, accurately performed, and clearly described. Results are convincingly discussed.
Therefore, in my opinion, the paper deserves publications.
However, more care should be paid to the language. It seems as the manuscript was not re-read with attention. 
Some observation are in listed in the following (but the list might be incomplete)

Line 51 hydroxybutyrate  better: hydroxybutanoate
Line 91 “Epoxidized”  why capital letter?
Line 147 “centrifuged that” ?  maybe “at”?
Figure 1: Maybe over the arrow the formula of maleic anhydride will be more “reader friendly” than C4H2O3
Lines 488-489
“Analyses of the morphological surface shows the appearance of spherical voids was evident morphological analyses of the breakage surfaces show spherical domains belonging to TPS, since it is partially miscible in PLA.”  I could not understand the sentence. Either something is missing or something was not deleted.
Lines 163 
“developed” does not seem the best verb. Maybe “obtained”?
Line 166 “zip bad”  ?  did you mean “bag”?
Line 236  plotted instead of ploted
Line 240 “with a temperature of 200 reports”  ?
Lines 240-243 Some mixed-up occurred, since the sentence “reports in the case of 240 vegetable oils the maleinization reaction requires at least temperatures close to 200 °C” is written twice and, as such, it makes no sense.
Line 246 “can attached”:  either ‘can attach’ or ‘can be attached’.
Line 248 adduct instead of “aduct”
Line 249 “molecule” here is better than mole
Line 272 “it is required the addition” better “the addition is required”
Line 287 “Different previous research, ….. have obtained similar values” the subject is plural maybe “studies” sounds better than “researchs”
Line 302 “Figure 4 a) are characteristics of ”  Figure 4a) is characteristic
Line 307 “responsible of”  responsible for
Line 340 “as well as”  ??  better “and”
Line 345 “As can be seen”  As it can be seen. The same is in Line 363
Line 442 “a noticeable visual change are detected.”  Either “a noticeable …. is” or “noticeable  ... .changes … are”
Line 446 “as has been summarized”  either “as it has been summarized” or, simply, “as summarized”
Line 455 “As has been mentioned above” either “As it has been mentioned above” or  “As above mentioned”
Line 458 “that de”  that the
Line 488 Analyses of the morphological surface shows : Analysis … shows  or “analyses show
“Analyses of the morphological surface shows the appearance of spherical voids was evident morphological analyses of the breakage surfaces show spherical domains belonging to TPS, since it is partially miscible in PLA.”  Sentence confused. Please, rephrase

Supporting Information

Maybe abbreviations should be listed in alphabetical order, so that they are easier to retrieve

“EKO    epoxidizedkaranja oil”     insert space, epoxidized karanja
KOH is a chemical formula (therefore self-explaining), not an abbreviation

Author Response

In the attached Word document, reviewer can be found a point-by-pint response to the reviewer´s comments. 

Reviewer 2 Report

This paper reports on the “Development of Poly(lactic-acid)-Thermoplastic Starch Formulations Using Maleinized Hemp Oil as Biobased Plasticizer”. The introduction, results and discussion are interesting and clearly presented. Description methodology and reference seems to be correct.

I have one comment to the manuscript:

  1. The tensile modulus should be named Modulus Young’s.

Taking into account all comments the manuscript may be published in Polymers after minor revision.

Author Response

In the attached Word document, reviewer can find a point-by-pint response to the reviewer´s comments. 
